# Effects of Phone-Based Psychological Intervention on Caregivers of Patients with Early-Onset Alzheimer’s Disease: A Six-Months Study during the COVID-19 Emergency in Italy

**DOI:** 10.3390/brainsci12030310

**Published:** 2022-02-24

**Authors:** Manuela De Stefano, Sabrina Esposito, Alessandro Iavarone, Michele Carpinelli Mazzi, Mattia Siciliano, Daniela Buonanno, Danilo Atripaldi, Francesca Trojsi, Gioacchino Tedeschi

**Affiliations:** 1First Division of Neurology, University of Campania “Luigi Vanvitelli”, 80138 Naples, Italy; manuela.destefano@policliniconapoli.it (M.D.S.); sabrina.esposito1@unicampania.it (S.E.); matsic@hotmail.it (M.S.); daniela.buonanno@policliniconapoli.it (D.B.); gioacchino.tedeschi@unicampania.it (G.T.); 2CDCD AORN “Ospedali dei Colli”—CTO, 80131 Naples, Italy; alessandro.iavarone@ospedalideicolli.it (A.I.); michelemm@hotmail.it (M.C.M.); 3Department of Advanced Medical and Surgical Sciences, University of Campania “Luigi Vanvitelli”, 80138 Naples, Italy; daniloatripaldi@hotmail.it; 4“Tieni A Mente—TAM” Association, San Giorgio a Cremano, 80046 Naples, Italy

**Keywords:** psychological intervention, EOAD, Alzheimer’s disease, caregiver, telemedicine, COVID-19

## Abstract

Caregivers of patients with early-onset Alzheimer’s disease (EOAD) experience higher level of burden, stress, and depression, due to premature role changes and social isolation. Moreover, the SARS-CoV-2 pandemic compelled restrictions regarding social interactions and mobility in Italy from March 2020, prompting telemedicine approaches for supporting patients and their families confined at home. We reported our experience regarding the effects of psychological phone-intervention (phone-I) on EOAD caregivers during pandemic. Twenty caregivers of EOAD patients were randomly assigned to treatment (TG) or control (CG) group. TG weekly underwent a phone-I for one month. All participants were assessed for caregiver burden and needs, anxiety and depression levels, and subjective impact of traumatic events at baseline (T0), at the fifth week (T1) and after 6 months (T2) from phone-I. We observed higher vulnerability to post-traumatic stress in TG compared to CG in all timepoints (*p* ≤ 0.05). Decreased stress effects and caregiver burden were revealed in TG at T1 compared to T0 (*p* ≤ 0.05), although showing an increase of these measures at T2 in the treated caregivers. Our findings suggest that although TG showed a peculiar vulnerability to post-traumatic stress, they showed increased wellbeing immediately after phone-I. However, this benefit disappeared six months later, along with the second infection wave, probably due to “exhaustion stage” achievement in “General Adaptation Syndrome”. This trend may suggest a beneficial but not solving role of a prompt phone-I on burden of caregivers of EOAD patients during the SARS-CoV-2 emergency.

## 1. Introduction

The ongoing pandemic of SARS-CoV-2 infection has caused over 358 million confirmed cases and over 5.54 million deaths, spreading from Asia to Europe since February 2020 (World Health Organization 2021). Its potential clinical manifestations immediately suggested high variability in symptoms and involved systems, showing worse prognosis in case of host comorbidities and increasing age [1]. Italy was the first European country to apply a nationwide lockdown from 9 March to 3 May 2020, compelling restrictive measures regarding mobility limitations, social distancing, and gradual suspension of any medical “non-essential” activity [2,3,4].

Older adults, especially those with cognitive impairment and assisted at home, are usually vulnerable to the effects of social distancing [5], routine disruption, as well as reduction of social and health supports at the community level [6].

Indeed, due to physical and social isolation associated to restrictive public health measures, a worsening in motor and cognitive functions and a significant increase of behavioral disturbs were observed in patients with dementia, aggravating functional limitations, caregiver’s burden, and the pre-existing conflicts within the family [7,8].

Generally, both formal and informal caregivers of people with dementia are expected to cope with disease symptoms and progression, enduring stress and burden. In particular, people with Alzheimer’s disease (AD) mainly depend on family caregivers for assistance, influencing negatively on psychological wellbeing through increased frequency of reporting mood disorders, risk of illness, accidents and death [9,10].

The early onset of dementia can exacerbate these aspects due to the premature impairment of family and social roles. Previous research showed that spouses of people with early onset AD (EOAD) precociously experienced negative changes due to responsibility shifting and social isolation [11]. Pressed by the dual care demand for both children and relatives, the members of the “sandwich generation” caregivers (i.e., the emerging cohort of caregivers, mainly represented by middle-aged women, who were caring for maturing children and aging relatives simultaneously) [12] frequently showed high level of burden, stress, and mood disorders, with family distress and higher risk of institutionalization [13], especially when behavioral-psychological symptoms occurred [14].

As part of the measures aimed at supporting the assistance of patients and caregivers in Italy, recent digital changes in health system prompted online care approaches, but the use of digital technologies in mental health and in the psychological field is recent and not still widespread [15]. In 2017, the National Council of Psychologists issued the recommendations for telepsychology providing guidelines for procedural changes. None online psychological practice was explicitly forbidden [16].

The pandemic led to a rapid implementation of several technology-based tools, to allow health care even in absence of physical contact, accelerating the switch from traditional “face-to-face” therapies to digital ones. Since the first lockdown, telemedicine and telepsychology were strongly encouraged, especially for managing frail people suffering from chronic diseases and their caregivers, in order to overcome specific care needs, organizational difficulties, geo-architectural barriers, and the interruption of formal assistance [17,18]. The use of phone-based methods in psychological counseling has a long history, particularly if related to prevention of suicidal crisis [19], and its specificity was widely examined [20]. Nevertheless, its role in managing psychosocial stress following community trauma or global-scale phenomena, such as the SARS-CoV-2 pandemic and its consequences, was not completely investigated [21].

Previous evidence underlined the positive effects of telephone support interventions specifically addressed to caregivers, reporting benefits for patients on physical, ecological, cognitive, and behavioral functions [22,23,24], and for caregivers on perceived emotional burden, degree of support and quality of the knowledge acquired [24,25]. On the other hand, satisfying relationships between caregivers and patients and a good family functioning were associated with a sense of competence, security and support, and general well-being [26], essential in critical and stressful situations such as a pandemic emergency.

On this background, the aim of our study was to explore the effect of a telephone psychological intervention (phone-I) on burden, mood disorders, needs, and post-traumatic symptoms of caregivers of patients with EOAD during the COVID-19 emergency.

## 2. Materials and Methods

Twenty consecutive caregivers of patients with EOAD (one each) were recruited at the First Division of Neurology of “Luigi Vanvitelli” University and AORN “Ospedali dei Colli”. The inclusion criteria were as follows: age > 18 years; being a family caregiver of a person with a diagnosis of EOAD [27]; spending at least four hours per day with the patient [28]; understanding the purpose of the study and signing the informed consent. Caregivers with communication and hearing problems, and/or inability to comply with the study commitments were excluded. The caregiver sample was matched by age, education level, and global cognitive score (Mini-mental state examination, MMSE) of patients with EOAD. Ten consenting caregivers were randomly assigned to the treatment (TG) and ten to the control (CG) group. The caregivers of TG underwent a phone-based psychological intervention (I) during the COVID-19 emergency, immediately preceded (T0) and followed (T1) by the administration of clinical scales (i.e., short-term assessment). The same scales were repeated six months after phone-I (T2, long-term assessment). The CG only performed the scales, at the same timepoints (T0, T1, T2). Furthermore, this study design aimed at exploring, in a real-life study, the psychological profile of the studied caregivers in two different social situations: T0 = immediately after Italian lockdown with mobility restrictions and social isolation and T2 = after the end of lockdown restrictions. The study lasted 6 months and was conducted in accordance with the Helsinki Declaration; informed consent was acquired from each participant by e-mail. The project was approved by the Institutional Review Board of the coordinating center (Ethics Committee of the University of Campania “L. Vanvitelli”—AORN “Ospedali dei Colli”).

### 2.1. Description of Phone-Intervention

The phone-I was conducted in May 2020 and consisted of 4 telephone sessions per participant, each lasting 60 min, once a week for 4 consecutive weeks, immediately after the first Italian lockdown, according to the guidelines of “counselling” formulated by American Psychological Association [29]. The calls were held in a comfortable environment, by just one licensed psychologist/psychotherapist, with a robust expertise in dementia and cognitive disorders. The approach was based on a nondirective control condition used by Borkovec and Costello for generalized anxiety disorder [30]. The primary goal was to provide non-directive support for caregivers through empathic/reflective listening and open-ended questioning. The first part of the phone-I was based on establishing a relationship with each caregiver, in order to allow free expression of relevant feelings regarding disease and care experience. The consults were focused on physical, cognitive, behavioral functioning, and daily routines of patients, on the perceived quality of relationship between patient and caregiver, on emotional, physical, and social burden perceived by caregivers, on significant needs, such as the spiritual one.

### 2.2. Clinical Assessment

The following scales were administered in Italian language to all caregivers at the three timepoints by the same licensed psychologist (M.D.S.):-The Caregiver Burden Inventory (CBI) [31]: a 24-item multi-dimensional questionnaire measuring caregiver burden with five subscales, namely “time dependence”, “developmental”, “physical”, “social”, and “emotional burden”. The score for each item is evaluated using a five-point Likert scale, ranging from 0 (not at all disruptive) to 4 (very disruptive) and all scores are summed; higher scores correspond to higher burden.-The Zung Self-Rating Anxiety Scale (SAS) [32]: a 20-item self-report assessment device used to measure anxiety levels, based on scoring in 4 symptom groups (cognitive, autonomic, motor, and central nervous system area). Each question is scored on a Likert scale of 1 (a little of the time)–4 (most of the time). Some questions are negatively worded to avoid the problem of set response. Total score ranges from 20 to 80.-The Zung Self-Rating Depression Scale (SDS) [33]: a self-administered survey, assessing the depressive status. Among 20 items, 10 are positively and 10 negatively worded questions. Each question is scored on a scale of 1 (a little of the time)–4 (most of the time), and the total score ranges from 20 to 80.-The Impact of Event Scale—Revised (IES-R) [34]: 22 questions which aim at measuring the subjective response to a specific traumatic event, through the response sets of intrusion (intrusive thoughts, feelings and imagery, nightmares, dissociative-like re-experiencing), avoidance (numbing of responsiveness, avoidance of feelings, situations, and ideas), and hyperarousal (anger, irritability, hypervigilance, concentration difficulty, heightened startle), as well as a total subjective stress IES-R score. Scores higher than 33 are associated to higher concern for post-traumatic stress disorder and to well-being impairment.-Caregiver Need Assessment (CNA) [35]: a 17 items tool referring to emotional, physical, functional, cognitive/behavioral, relational, social/organizational, and spiritual needs. These topics were grouped into two interest areas: need of emotional/social support; need of information/communication, with a total score ranging from 0 to 51, directly proportional to the perceived intensity of need.

### 2.3. Statistical Analysis

All data were tested for normality via the Kolmogorov-Smirnov (K-S) test, and the departure from normality of T1 CNA (K-S = 0.269, *p* = 0.039), T1 NPI (K-S = 0.272, *p* = 0.035), and T2 CBI (K-S = 0.269, *p* = 0.039) oriented for a non-parametric statistical approach. Between-group comparisons were performed by the Mann-Whitney U test. Changes of assessment score (i.e., within-group comparison) were checked by the Wilcoxon rank test. All multiple and pairwise post-hoc comparisons were corrected for Benjamini-Hochberg procedures; Benjamini-Hochberg corrected *p*-value ≤ 0.05 were considered statistically significant [36]. All analyses were performed using IBM Statistical Package for Social Science (SPSS) version 20.

## 3. Results

The study sample consisted of 20 age- and education-matched caregivers (respectively *p* = 0.173; *p* = 0.09) randomized into two groups, mainly spouses (60%) followed by children (30%) and siblings (10%), with a mean (M) age of 53 years, and with an average of 11.7 years of education.

Their age- and education-matched relatives (respectively *p* = 0.074; *p* = 0.094), homogeneously affected by moderate EOAD (*p* = 0.677; corrected MMSE M score = 14.9) were also middle-aged (M 59 years) and showed the same education level of the studied sample. (Clinical-demographic characteristics of study sample and of the EOAD patients are summarized in Table 1).

Between-group comparisons of the psychometric results (Table 2) showed differences in the IES-R scores, revealing higher scores in TG compared to CG in all three timepoints (T0, T1, T2).

Non-parametric Wilcoxon test within TG (Table 3) showed a significant decrease in post-traumatic stress (IES-R, *p* = 0.049) and burden (CBI, *p* = 0.011) of caregivers at T1 compared to these scores at baseline. Moreover, a trend toward reduction of depressive symptoms (Zung SDS) at T1 was observed as a rather positive effect of the phone-I on TG. However, TG showed a significant increase in post-traumatic stress symptoms (*p* = 0.008) and burden (*p* = 0.002) by comparing the scores at T1 to those at T2. Surprisingly, the CBI of TG was significantly higher at T2, especially when compared to the scores at T0 (*p* = 0.025). No other significant comparisons were observed in TG.

Regarding CG, non-parametric Wilcoxon test didn’t show significant differences in any of the scales administered over time (Table 4).

## 4. Discussion

In the context of a strict lockdown period due to the SARS-CoV-2 pandemic, our study aimed at investigating, for the first time, post-traumatic stress symptoms, needs, burden, depression, and anxiety at short-term and long-term timepoints after a psychological phone-I in a small sample of Italian familial caregivers of people with EOAD. After the 4-week phone-I, performed in May 2020, the “treated” caregivers showed a rather positive response in term of reduction in perceived care burden and post-traumatic stress symptoms, and a modest benefit on depression compared to the “untreated” caregivers, although reporting extinction of the benefit after 6 months.

Given the early and often drastic change in life habits of people with EOAD, all family members are directly or indirectly affected by diagnosis. Spouses and adult children are usually the main providers of informal caregiving [37]. In order to face the growing care demand, due to premature cognitive and functional impairment, caregivers may gradually neglect their own needs [37]. Frequently, the precocious disease onset and its faster progression induce a reversal in roles, with spouses and children becoming “the parents” of their own ill relatives, even at a relatively young age [38]. Additionally, caregiving people with EOAD, when compared to the same activity aimed at managing patients with late onset dementia (LOD), are usually longer lasting, with less social support, higher degree of burden, and more symptoms of depression [39]. Moreover, as compared to older caregivers, the younger ones are less able to accept the disease and its negative effects [40].

With the occurrence of the SARS-CoV-2 pandemic and following restrictive measures provided by governments, particularly the lockdown consisting in home confinement and reduction of national healthcare services, familial caregivers of patients with EOAD had to face new challenges also due to forced co-habitation and extreme uncertainty about the future. However, even if the number of support groups for caregivers and individuals with dementia is progressively growing, these services are still scarce and often designed for people with LOD [41].

Compared to CG, the TG showed a worse perception of “COVID-19 as traumatic event” at all timepoints (T0, T1, T2) highlighting even before the phone-I a greater vulnerability to symptoms of post-traumatic stress disorder then the CG. In our opinion, this random occurrence, together with the small sample size, may have influenced all results of the between-group analyses. Nevertheless, when performed the within-group analyses, the TG showed a significant improvement in post-traumatic stress symptoms after the phone-I (T1), reporting extinction of the benefit after 6 months (T2). Contrariwise, the CG showed no changes across the 3 timepoints. TG caregivers showed a significant reduction in the perceived care burden after the phone-I. Surprisingly, an increase in caregiver burden was observed in TG after 6 months from the phone-I, without revealing changes in neuropsychiatric symptoms of the cared patients. Furthermore, depressive symptoms also tended to decrease in TG after the phone-I, although this evidence was only a trend toward statistical significance. These results all together, although related to a small sample of subjects, could suggest a beneficial role of weekly phone-I on middle-aged caregivers of EOAD patients. In particular, this phone-I was based on empathic listening and open-ended questioning, specifically focused on their daily needs, perceived quality of relationship with patient, physical and social burden, and perception of patients’ disease symptoms. A similar approach could be useful in crisis periods, such as during SARS-CoV-2 pandemic, as well as routinely. Nevertheless, although in line with previous evidence showing the efficacy of telephone use in counselling and crisis intervention [19,20], especially in specific treatment modalities such as Cognitive Behavioural Therapy [42,43], Dialectal Behaviour Therapy [44,45] and psychoanalysis [46,47], our findings should be contextualized in their specific time frame. The phone-I was performed in May 2020, corresponding to the final phase of a strict limitation period (first Italian lockdown), while the T2 follow-up was performed in December 2020, when unexpectedly Italian people had newly to face with national restrictive measures to contain a second infection wave. Despite an initial greater vulnerability to stress of TG caregivers compared to CG, TG showed a significant reduction of their post-traumatic stress symptoms and burden after the phone-I, supporting the hypothesis of a positive influence of the phone-I on the TG. However, 6 months after the phone-I, in association with worsening of the pandemic course, the increase of CBI scores of TG, even higher than the baseline ones, could be explained as a “rebound effect” due to achievement of the “exhaustion stage” in the framework of the “General Adaptation Syndrome” [48]. Probably, the persistence of a stressor would have depleted physical and mental resources in TG caregivers who were more sensitive to traumatic events related to pandemic. In particular, the overall higher prevalence of post-traumatic stress symptoms in TG caregivers, associated with enduring and unresolved stressors, and depleting physical, emotional, and mental resources, hindered our study sample from coping with stress and caregiving-strain faced in T2.

Our study has several limitations. First, despite the randomization, the small sample size was not representative of the whole caregiver community of EOAD people, making our observations purely speculative. In particular, we noted higher scores at the IES-R in TG group already at T0, probably due to the randomization. This could have produced a transfer effect on the following evaluations that (at least in part) accounted for the observed differences. Second, we cannot exclude the effect of other confounding environmental factors, related to the historical period, on our results. However, the adopted tools assessed several psychological, behavioral and burden dimensions and this may have led us to rule out possible “hidden” variables exerting a common influence of such dimensions. Moreover, the potential beneficial effect of the phone-I on caregivers, related to the perception of being listened to and engaged in a supportive relationship, appear to be not long lasting, although effective, in the most emotionally vulnerable subjects, at least in the early stages of a traumatic experience. Probably, in order to strengthen the effectiveness of a phone-based psychological intervention, treatment regimen should be longer, tailored to specific caregiver categories, and focused on family relationship functioning, thereby increasing the sense of competence, essential in both ordinary and exceptional stressful situations.

## 5. Conclusions

In conclusion, our findings revealed that a phone-I targeted to caregivers of patients with EOAD might have short-term beneficial effect on stress-related and depressive symptoms in the most emotionally vulnerable subjects, at least in the early stages of a traumatic experience. A prompt psychological intervention, even phone-based, might be considered to mitigate but not to prevent the re-emerging symptoms related to a persistent traumatic event, such as a pandemic emergency. Further studies, such as larger randomized controlled trials, combining phone-based psychological interventions to face-to-face ones, will be useful to verify the benefits of these approaches. In particular, treatment regimen should be longer and more tailored to specific caregiver categories.

## Figures and Tables

**Table 1 brainsci-12-00310-t001:** Descriptive statistics of caregivers and EOAD patients.

Variable	Control Group Mean (SD)	Treatment Group Mean (SD)	*p*
Caregivers’ age, years	57.7 (7.7)	49 (14.9)	0.173
Caregivers’ years of education	13.5 (4.1)	10.0 (4.3)	0.090
Patients’ age, years	61.0 (5.0)	57.6 (3.8)	0.074
Patients’ years of education	13.5 (4.1)	9.9 (4.2)	0.094
Patients’ MMSE corrected scores	15.3 (5.6)	14.4 (6.8)	0.677

Note. SD, Standard Deviation; MMSE, Mini Mental State Examination.

**Table 2 brainsci-12-00310-t002:** Between-group analysis at the three timepoints.

Scale	*p* (TG vs. CG) T0	*p* (TG vs. CG) T1	*p* (TG vs. CG) T2
Zung SDS	0.94	0.16	0.65
Zung SAS	0.34	0.88	0.29
CNA	0.73	0.57	0.57
CBI	0.73	0.97	0.082
IES-R	0.008 *	0.05 *	0.049 *
NPI (patients)	0.65	0.38	0.91

CAN, Caregiver Need Assessment; CBI, Caregiver Burden Inventory; IES-R, Impact of Event Scale—Revised; NPI, Neuropsychiatric Inventory; Zung SAS, Zung Self-Rating Anxiety Scale; Zung SDS, Zung Self-Rating Depression Scale; statistically significant differences are marked by *.

**Table 3 brainsci-12-00310-t003:** Longitudinal assessment of measures of caregivers’ depression, anxiety, needs, burden, post-traumatic stressand neuropsychiatric symptoms in the TG.

Scale	T0 M (±SD) [CI]	T1 M (±SD) [CI]	T0 M (±SD) [CI]	T0 vs. T1	T1 vs. T2	T0 vs. T2
Zung SDS	49.90 (9.71) [43.88–55.92]	46.40 (9.17) [40.72–52.08]	51.30 (6.39) [47.34–55.26]	0.065	0.066	0.767
Zung SAS	54.33 (5.03) [51.21–57.45]	42.20 (7.13) [37.78–46.72]	51.00 (5.29) [47.72–54.28]	0.285	0.066	0.172
CNA	31.10 (8.56) [25.79–36.41]	32.70 (9.82) [26.71–38.79]	49.70 (16.73) [39.33–60.07]	0.759	0.126	0.683
CBI	36.40 (18.37) [25.01–47.79]	24.00 (16.35) [13.87–34.13]	49.70 (16.73) [39.33–70.07]	0.011 *	0.002 *	0.025 *
IES-R	44.00 (13.94) [35.36–52.64]	32.10 (8.62) [26.66–37.44]	53.40 (11.32) [46.38–60.42]	0.049 *	0.008 *	0.221
NPI	34.10 (16.16) [24.08–44.12]	39.60 (13.47) [31.25–47.25]	46.67 (4.16) [44.09–49.25]	0.063	0.593	0.683

CAN, Caregiver Need Assessment; CBI, Caregiver Burden Inventory; IES-R, Impact of Event Scale—Revised; NPI, Neuropsychiatric Inventory; Zung SAS, Zung Self-Rating Anxiety Scale; Zung SDS, Zung Self-Rating Depression Scale; statistically significant differences are marked by *.

**Table 4 brainsci-12-00310-t004:** Longitudinal assessment of measures of caregivers’ depression, anxiety, needs, burden, post-traumatic stressand neuropsychiatric symptoms in the CG.

Scale	T0 M (±SD) [CI]	T1 M (±SD) [CI]	T0 M (±SD) [CI]	T0 vs. T1	T1 vs. T2	T0 vs. T2
Zung SDS	48.6 (2.87) [46.82–50.38]	49.5 (3.75) [47.18–51.82]	49.5 (12.23) [41.92–57.08]	0.398	0.592	0.878
Zung SAS	41.1 (11.4) [34.03–48.17]	42.9 (10.22) [36.57–49.23]	42.8 (12.80) [34.87–50.73]	0.094	1.00	0.593
CNA	33.3 (13.61) [24.87–41.74]	35.4 (10.92) [28.63–42.17]	34.2 (10.88) [27.46–40.94]	0.767	0.722	0.919
CBI	32.5 (27.82) [15.26–49.74]	29.9 (24.97) [14.42–45.38]	31 (22.19) [17.25–44.75]	0.262	0.959	0.514
IES-R	20.5 (16.26) [10.42–30.58]	20.4 (19.89) [8.07–32.73]	28.9 (25.77) [12.99–44.81]	0.953	0.123	0.155
NPI	35.5 (13.18) [27.33–43.67]	34.6 (13.78) [26.06–43.14]	34.5 (11.99) [27.07–41.93]	0.677	0.953	0.539

CAN, Caregiver Need Assessment; CBI, Caregiver Burden Inventory; IES-R, Impact of Event Scale—Revised; NPI, Neuropsychiatric Inventory; Zung SAS, Zung Self-Rating Anxiety Scale; Zung SDS, Zung Self-Rating Depression Scale.

## Data Availability

The spreadsheet database of the cases is available upon request to the corresponding author.

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
