# Peer review of "Effects of Phone-Based Psychological Intervention on Caregivers of Patients with Early-Onset Alzheimer’s Disease: A Six-Months Study during the COVID-19 Emergency in Italy"

_brainsci, 2022, doi:10.3390/brainsci12030310_

Round 1

Reviewer 1 Report

1) The title should mention the article type and where the manuscript was done.

2) Abstract should be revised. Structure, methodology, and results should be addressed.

3) It is advised English grammatical corrections throughout the text. Punctuation, spaces.

4) Please, remove grouped references in all sections. ‘‘Previous evidence underlined the positive effects of telephone support interventions specifically addressed to caregivers, [23-26] with benefits for both patients, on physical, ecological, cognitive, and behavioral functions [27,28], and caregivers on perceived emotional burden [28,29], degree of support and quality of the knowledge acquired [30,31].’’

5) Methods

a) Why reference 33 was cited in this phrase ‘‘being a family caregiver of a person 95 with a diagnosis of EOAD [33]’’?

b) From where was chosen this amount of time ‘‘spending at least four hours per day with the patient’’?

c) Why these times were chosen T0, T1, and T2?

d) Do the applicants of the scales have permission for their application? In what language were the scales provided?

e) Statistical analysis should be better described. What were the statistical methods applied? Data distribution?

6) Results

a) Tables 1, 2, and 3 could be summarized in only one table.

7) Could the authors provide the spreadsheet database of the cases as supplementary material?

8) How did the authors exclude confounding factors?

IDEAS

Provide a figure about the possible explanation of the findings in the present study. A table with a comparison of mainly described psychological factors throughout the pandemic.

Author Response

Comments and Suggestions for Authors

Reviewer#1

1) The title should mention the article type and where the manuscript was done.

Authors: We thank the Reviewer#1 for this valuable suggestion. We modified the title as follows: “Longitudinal study of effects of phone-based psychological intervention on caregivers of patients with early-onset Alzheimer’s disease during the Covid-19 emergency in Italy”.

2) Abstract should be revised. Structure, methodology, and results should be addressed.

Authors: According to the Reviwer#1’s suggestion, we substantially rewrote the abstract, clarifying methods and results (lines 15-29). Moreover, according to a Reviewer#2’s comment, we deleted some abbreviations to make the abstract easier to read (please see Reviewer#2’s comments).

3) It is advised English grammatical corrections throughout the text. Punctuation, spaces.

Authors: We thank for this valuable suggestion and apologize for grammatical and typographical inaccuracies and corrected them throughout the text (i.e., lines 38-40, 41, 43-46, 48, 51, 52, 56, 57, 70-72, 76, 77, 83-87, 91, 93, 96, 97, 103-105, 146, 147, 149, 150, 151, 157, 158, 175, 176, 177-182, 184-188, 190-192, 194, 202-206, 209-212, 214-221, 230-233, 235, 236, 241, 242, 244, 247-250, 253, 254, 282, 302).

4) Please, remove grouped references in all sections. ‘‘Previous evidence underlined the positive effects of telephone support interventions specifically addressed to caregivers, [23-26] with benefits for both patients, on physical, ecological, cognitive, and behavioral functions [27,28], and caregivers on perceived emotional burden [28,29], degree of support and quality of the knowledge acquired [30,31].’’

Authors: According to this suggestion, we removed grouped references (in all sections) where inappropriate (lines 45, 86, 87).

5) Methods

  1. a) Why reference 33 was cited in this phrase ‘‘being a family caregiver of a person 95 with a diagnosis of EOAD [33]’’?

Authors: We thank the Reviewer#1 for the opportunity to clarify the use of reference 33 (line 99): the authors extensively reviewed the definition and the peculiar phenomenon of young-onset dementia. Rossor et al. described both epidemiological and clinical features, also reporting the differences with late-onset dementia. Our study sample, made up of caregivers of patients with young-onset Alzheimer's disease, had to face very peculiar issues due to the early diagnosis. Therefore, we considered appropriate to provide an overview of its distinctive features in order to outline the specific background of the sample.

  1. b) From where was chosen this amount of time ‘‘spending at least four hours per day with the patient’’?

Authors: We thank the Reviewer#1 for the requested clarification about the debated definition of "informal/family caregiver". Time dedicated to caregiving is expected to vary between settings, depending on dementia severity, and social expectations. The amount of time spent caring is ranging from 50 to 286 hours per month [Brodaty H, Drugs Aging. 2002]. Furthermore, as the disease progresses the amount of time increases with costs in term of burden for caregivers. In a Swedish study, caregiving was divided into surveillance/supervision, help in activities of daily living (ADL), instrumental activities (IADL) and other tasks. Average help for ADL was 2.5 hours per day, compared to about twice for IADL (4.8 hours) [Wimo A, Int J Geriatr Psychiatry 2000]. Some recent phase 3 RCT (e.g.  BAN2401, etc.) required as caregiver’s inclusion criteria a minimum of 8 hours per week to spent with the patient. In our study, we restricted participation to “primary” family caregiver adopting a time threshold (4 diurnal hours per day), which ensured a significant proximity to the patient. We added the reference 34 (Wimo et al., 2000).

  1. c) Why these times were chosen T0, T1, and T2?

Authors: We thank for this request giving us the opportunity to clarify the study design. We administered the scales in the week preceding the beginning of phone intervention (T0) with the aim of drawing the baseline psychological profiles of all caregivers involved in the study. Then, we re-evaluated both study arms in the week immediately after the end of phone intervention (T1, short-term evaluation), to explore the effects of treatment on scales i) over time within the active group, ii) and in comparison with the non-interventional group.

T2 was set up 6 months after the end of phone-intervention, to verify its potential long-term effect on scales in treatment group. This study design aimed at exploring the psychological profile of the studied caregivers in two different social situations: T0 = immediately after Italian lockdown with mobility restrictions and social isolation and T2 = after the end of lockdown restrictions. However, being a real-life study, in T2 (December 2020) the social conditions resulted very similar to the baseline showing us a "rebound effect" of the scores as reported in the discussion (line 272-273). We clarified the study design according to this Reviewer#1’s comment (lines 107-112).

  1. d) Do the applicants of the scales have permission for their application? In what language were the scales provided?

Authors: We thank the Reviewer#1 for having request this clarification. All the scales used were administered in Italian language by a licensed neuropsychologist / psychotherapist with over 10 years' experience in the management of dementia patients and their caregivers. We underlined thee aspects in the revised version (lines 135-136). Furthermore, all scales have been validated in Italian language [Marvardi, M. Aging Clin Exp Res, 2005; Craparo, G. Neuropsych Dis Treat, 2013; Moroni, L.; G Ital Med Lav Ergon 2008; Innamorati, M. Psicoter. Cogn. e Comportamentale 2006], except for the Zung SAS, and are available online (except for the CNA which was directly provided by the author). These scales have been widely used for investigating mood, burden and needs of caregivers of patients with several neurological disorders [Altieri, M. Am J Geriat Psychiat 2021; Maggio, M. G. J Clin Med 2021; Zammitti, A. Eur J Investig Health Psychol Educ 2021; Brivio, E. BMC Public Health 2021; Graffigna, G. BMC Health Serv Res 2021; Cova, I. Neurol Sci 2018; Carpinelli Mazzi, M. Eur Geriatr Med 2020; Lasalvia, A. Fam pract 2022; Grandinetti, P. Brain sci 2021].

  1. e) Statistical analysis should be better described. What were the statistical methods applied? Data distribution?

Authors: According to these suggestions, we have modified the description of statistical analysis as follows (paragraph 2.3, lines 165-170): “All data were tested for normality. Between-group comparisons were per-formed by the Mann-Whitney U test. Changes of assessment score (i.e., within-group comparison) were checked by the Wilcoxon rank test. All multiple and pairwise post-hoc comparisons were corrected for Benjamini-Hochberg procedures; Benjamini-Hochberg corrected p-value ≤ 0.05 were considered statistically significant (Benjamini and Hochberg, 1995). All analyses were performed using IBM Statistical Package for Social Science (SPSS) version 20.”.

6) Results

  1. a) Tables 1, 2, and 3 could be summarized in only one table.

Authors: We thank the Reviewer#1 for this suggestion: we merged tables 1 - 3 into the revised “Table 1”, included in the revised text (lines 183-184).

Table 1. Descriptive statistics of caregivers and EOAD patients.

Variable

Control group

Mean (SD)

Treatment group

Mean (SD)

p

Caregivers’ age, years

57.7 (7.7)

49 (14.9)

0.173

Caregivers’ years of education

13.5 (4.1)

10.0 (4.3)

0.090

Patients’ age, years

61.0 (5.0)

57.6 (3.8)

0.074

Patients’ years of education

13.5 (4.1)

9.9 (4.2)

0.094

Patients’ MMSE corrected scores

15.3 (5.6)

14.4 (6.8)

0.677

Note. SD, Standard Deviation; MMSE, Mini Mental State Examination.

7) Could the authors provide the spreadsheet database of the cases as supplementary material?

Authors: We thank the Reviewer#1 for having raised this critical aspect. We did not include the original datasheet as supplementary material. However, we specified in the revised text “the spreadsheet database of the cases is available upon request to the corresponding author”.

8) How did the authors exclude confounding factors?

Authors: We thank the Reviewer#1 for having raised this further critical aspect. We acknowledged in the limits section that the role of potential confounding environmental factors, related to the historical period, could not be excluded (lines 284-288). However, the adopted tools assessed several psychological, behavioral and burden dimensions and this may have led us to rule out possible “hidden” variables exerting a common influence of such dimensions.  

IDEAS

Provide a figure about the possible explanation of the findings in the present study. A table with a comparison of mainly described psychological factors throughout the pandemic.

Authors: We appreciate this valuable suggestion of the Reviewer#1 but, given the limited time available to complete the revisions, we are unable to catch this interesting cue.

Reviewer 2 Report

Overall this is a well written paper which reports the study findings well. I have some fairly minor points which I think would be worth addressing:

Abstract:

First sentence of the abstract should be restructured to read ‘Caregivers of people with early-onset Alzheimer’s disease…’ and this wording should be tended to throughout

There are a lot of acronyms in the abstracts. These should be minimised as far as possible to increase readability. IES-R and the clinical scales are used without explanation. Better to tell us what a higher IES-R score means.

Background:

Please check sentence on lines 67/68 page 2. This reads a little oddly, please just check this is the correct information. In 2017, the National Council of Psychologists issued the recommendations for telepsychology without discouraging any online psychological practices [17].

Methods:

The scales chosen appear appropriate. However as a qualitative researcher I do not feel I can comment further on the accuracy of the statistical results.

I would be interested to know why the intervention was set up in the way that it was? Why 4 weeks of phone calls and why consecutive weeks? And then why follow-up at 6 months? Is it possible to say?

Discussion:

Generally this is well written. Importantly the authors pick up on the timelines within which the study was carried out and the potential impacts of these events on the findings. They also report negative and neutral findings which good to see.

I think the discussion would benefit from exploring the nature of the intervention a little more and how this could be adapted in the future – this is briefly stated in the conclusion but going forward these are important points and I think they warrants a little development here. While the authors note the small sample size, it would also be worth noting that a brief qualitative element may have given light to some of their more unexpected findings.

Suggest restructuring this a little to start with a summary of key findings before moving into the synthesis with the current literature.

Conclusion:

In this section important points are made about the length of the intervention and the potential to tailor this to family caregivers. This reads well

Author Response

Reviewer#2

Overall this is a well written paper which reports the study findings well. I have some fairly minor points which I think would be worth addressing:

Authors: We thank the Reviewer for her/his valuable comments.

Abstract: First sentence of the abstract should be restructured to read ‘Caregivers of people with early-onset Alzheimer’s disease…’ and this wording should be tended to throughout

Authors: We modified the first sentence, as well as the same wording throughout the text, also according to the second and third comments of Reviewer#1.

There are a lot of acronyms in the abstracts. These should be minimized as far as possible to increase readability. IES-R and the clinical scales are used without explanation. Better to tell us what a higher IES-R score means.

Authors: According to this valuable comment, we minimized the use of acronyms in the abstract, explicating what the used neuropsychological scales mean. Moreover, the abstract was substantially revised also according to the second comment of the Reviewer#1.

Background:

Please check sentence on lines 67/68 page 2. This reads a little oddly, please just check this is the correct information. In 2017, the National Council of Psychologists issued the recommendations for telepsychology without discouraging any online psychological practices [17].

Authors: we thank the Reviewer#2 for this suggestion and reformulated the sentence as follows: “In 2017, the National Council of Psychologists issued the recommendations for telepsychology providing guidelines for procedural changes. None online psychological practice was explicitly forbidden [17]” (lines 71, 72).

Methods:

The scales chosen appear appropriate. However as a qualitative researcher I do not feel I can comment further on the accuracy of the statistical results.

Authors: We thank the Reviewer#2 for the valuable appreciation. However, also according to the fifth comment of the Reviewer#1, we clarified the methods section (lines 165-170).

I would be interested to know why the intervention was set up in the way that it was? Why 4 weeks of phone calls and why consecutive weeks? And then why follow-up at 6 months? Is it possible to say?

Authors: We thank for these comments that permit us to clarify the guidelines of “counselling” which we have fulfilled. According to American Psychological Association, counselling psychology is typically held once a week for 4 weeks and is focused on “emotional, social, work, school, and physical health concerns people may have at different stages in their lives, on typical life stresses and on severe issues with which people may struggle as individuals and as a part of families, groups and organizations” (Consensus Conference 2020). The approach we used in our study was based on a nondirective control condition used by Borkovec and Costello (J. Consult. Clin. Psychol. 1993, ref. 36) with the aim of providing non-directive support for caregivers through empathic / reflective listening and open-ended questioning. We briefly underlined these aspects in the text. Moreover, also according to the fifth comment of the Reviewer#1 (c), we clarified the study design that aimed at exploring the psychological profile of the studied caregivers in two different social situations using short-term and long-term assessments (please see also the response to the fifth comment of Reviewer#1).

Discussion:

Generally this is well written. Importantly the authors pick up on the timelines within which the study was carried out and the potential impacts of these events on the findings. They also report negative and neutral findings which good to see.

Authors: We thank the Reviewer#2 for her/his accurate analysis of the different sections of the paper.

I think the discussion would benefit from exploring the nature of the intervention a little more and how this could be adapted in the future – this is briefly stated in the conclusion but going forward these are important points and I think they warrants a little development here. While the authors note the small sample size, it would also be worth noting that a brief qualitative element may have given light to some of their more unexpected findings.

Authors: According to this valuable suggestion, we emphasized in the last paragraph of the discussion the qualitative aspects of the performed psychological intervention that could be helpful in the future in order to better support familial caregivers of patients with EOAD (lines 288-295).

Suggest restructuring this a little to start with a summary of key findings before moving into the synthesis with the current literature.

Authors: According to this helpful comment, we restructured the starting paragraphs of the discussion and of the conclusions reporting a brief summary of the key findings (lines 214-221, 297-300).

Conclusion:

In this section important points are made about the length of the intervention and the potential to tailor this to family caregivers. This reads well.

Authors: According to this suggestion, we emphasized these points in the conclusions (lines 302-305).

Round 2

Reviewer 1 Report

  1. There are still grouped references in the manuscript that should be addressed. It is advised to reduce the number of references. In the majority of the journals is between 30-50. The present manuscript has 57 references.
  2. The title still needs to be revised. ‘‘Longitudinal studies employ continuous or repeated measures to follow particular individuals over prolonged periods of time—often years or decades.’’ Caruana EJ, Roman M, Hernández-Sánchez J, Solli P. Longitudinal studies. J Thorac Dis. 2015 Nov;7(11):E537-40. doi: 10.3978/j.issn.2072-1439.2015.10.63. PMID: 26716051; PMCID: PMC4669300.
  1. Statistical analysis:

It is advised professional statistical service to assess description.

a) Why Did the authors use the Mann-Whitney U test and Wilcoxon rank test if the distribution was normal?

b) Why Did you chose Benjamini-Hochberg test?

c) Please display confidence intervals in your tables. Also, present in your table significant results with (*)

d) Why did you not perform an ANOVA or a multiple regression?

Author Response

  1. There are still grouped references in the manuscript that should be addressed. It is advised to reduce the number of references. In the majority of the journals is between 30-50. The present manuscript has 57 references.

Authors: We apologize for the numerous and grouped references. We reduced the number of references to 48.

  1. The title still needs to be revised. ‘‘Longitudinal studies employ continuous or repeated measures to follow particular individuals over prolonged periods of time—often years or decades.’’ Caruana EJ, Roman M, Hernández-Sánchez J, Solli P. Longitudinal studies. J Thorac Dis. 2015 Nov;7(11):E537-40. doi: 10.3978/j.issn.2072-1439.2015.10.63. PMID: 26716051; PMCID: PMC4669300.

Authors: According to Reviewer#1’s suggestion, we still revised the title: “Effects of phone-based psychological intervention on caregivers of patients with early-onset Alzheimer’s disease: a six-months study during the Covid-19 emergency in Italy”.

  1. Statistical analysis:

It is advised professional statistical service to assess description.

  1. a) Why Did the authors use the Mann-Whitney U test and Wilcoxon rank test if the distribution was normal?

Authors: We thank the Reviewer#1 for these further advices. As suggested, we revised the description of statistical analysis according to the advices of a professional statistics service.

We apologize to the Reviewer for the unclear sentence concerning the test used to verify the distribution of the outcomes. In the revised version of the manuscript, we specified that the Kolmogorov-Smirnov (K-S) test was used to verify whether the distribution of outcomes deviated from comparable normal distributions. In the table below, we reported the results of the K-S tests, which suggested that the distributions of T1 CNA (K-S= 0.269, p= 0.039), T1 NPI (K-S= 0.272, p= 0.035), and T2 CBI (K-S= 0.269, p= 0.039) were significantly different from the normality. Therefore, we used the non-parametric tests which do not make a ‘normality assumption’ about the distribution of the outcomes. Particularly, in the revised version of the manuscript, we reported as follows: “All data were tested for normality via the Kolmogorov-Smirnov (K-S) test, and the departure from normality of T1 CNA (K-S= 0.269, p= 0.039), T1 NPI (K-S= 0.272, p= 0.035), and T2 CBI (K-S= 0.269, p= 0.039) oriented for a non-parametric statistical approach” (lines 165-167).

Kolmogorov-Smirnov (K-S) test

p-value

Baseline:

Zung SDS

0.200

0.200

Zung SAS

0.204

0.200

CNA

0.150

0.200

CBI

0.199

0.200

IES-R

0.157

0.200

NPI (patients)

0.153

0.200

Timepoint 1:

Zung SDS

0.188

0.200

Zung SAS

0.167

0.200

CNA

0.269

0.039

CBI

0.141

0.200

IES-R

0.160

0.200

NPI (patients)

0.272

0.035

Timepoint 2:

Zung SDS

0.181

0.200

Zung SAS

0.133

0.200

CNA

0.107

0.200

CBI

0.269

0.039

IES-R

0.156

0.200

NPI (patients)

0.146

0.200

Note. The variables with not normal distribution are signed in bold.

  1. b) Why Did you choose Benjamini-Hochberg test?

Authors: Considering the explorative nature of our study, we chose to use the Benjamini-Hochberg procedure to control for the false discovery rate, i.e. the proportion of results that were false positives.

  1. c) Please display confidence intervals in your tables. Also, present in your table significant results with (*)

Authors: We thank the Reviewer#1 for this valuable suggestion. We displayed confidence intervals in tables 3 and 4. Moreover, we marked significant results with (*).

  1. d) Why did you not perform an ANOVA or a multiple regression?

Authors: As reported in response to Comment 1, some variables were not normally distributed, therefore we used a non-parametric approach to prevent any inferential bias.

Round 3

Reviewer 1 Report

Satisfactory.